# The Influence of Strain and Sex on High Fat Diet-Associated Alterations of Dopamine Neurochemistry in Mice

**DOI:** 10.3390/nu16193301

**Published:** 2024-09-29

**Authors:** Kristen A. Hagarty-Waite, Heather A. Emmons, Steve C. Fordahl, Keith M. Erikson

**Affiliations:** Department of Nutrition, University of North Carolina at Greensboro, Greensboro, NC 27412, USA; kahagarty@uncg.edu (K.A.H.-W.); haemmons@uncg.edu (H.A.E.); scfordah@uncg.edu (S.C.F.)

**Keywords:** saturated fat, insulin resistance, voltammetry, anxiety-like behavior, microglial activation

## Abstract

**Objective:** The objective of this study was to determine the influence of sex and strain on striatal and nucleus accumbens dopamine neurochemistry and dopamine-related behavior due to a high-saturated-fat diet (HFD). **Methods:** Male and female C57B6/J (B6J) and Balb/cJ (Balb/c) mice were randomly assigned to a control-fat diet (CFD) containing 10% kcal fat/g or a mineral-matched HFD containing 60% kcal fat/g for 12 weeks. **Results:** Intraperitoneal glucose tolerance testing (IPGTT) and elevated plus maze experiments (EPM) confirmed that an HFD produced marked blunting of glucose clearance and increased anxiety-like behavior, respectively, in male and female B6J mice. Electrically evoked dopamine release in the striatum and reuptake in the nucleus accumbens (NAc), as measured by ex vivo fast scan cyclic voltammetry, was reduced for HFD-fed B6J females. Impairment in glucose metabolism explained HFD-induced changes in dopamine neurochemistry for B6J males and, to a lesser extent, Balb/c males. The relative expressions of protein markers associated with the activation of microglia, ionized calcium binding adaptor molecule (Iba1) and cluster of differentiation molecule 11b (CD11b) in the striatum were increased due to an HFD for B6J males but were unchanged or decreased amongst HFD-fed Balb/c mice. **Conclusions:** Our findings demonstrate that strain and sex influence the insulin- and microglia-dependent mechanisms of alterations to dopamine neurochemistry and associated behavior due to an HFD.

## 1. Introduction

The chronic intake of a high-saturated-fat diet (HFD) leads to metabolic and neurobiological consequences such as insulin resistance, the activation of microglia, and the dysregulation of the dopaminergic system [1,2]. Diet-induced obesity has been demonstrated to blunt striatal dopamine responses to highly palatable food, leading to changes in food and reward-seeking behavior in human subjects [3]. HFD-induced mechanisms that contribute to altered dopamine neurochemistry include dysregulated maximal dopamine release and reuptake through the dopamine transporter (DAT) in dopamine-rich brain regions like the striatum and nucleus accumbens (NAc) [4,5,6,7,8]. Reduced striatal dopamine reuptake and integration of DAT into synaptosomal membranes have been shown following as few as 6 weeks of HFD feeding [5]. Longer-term studies have demonstrated the persistent impacts of an HFD on DAT recruitment and function, as well as changes to dopamine receptor expression, ultimately leading to behavioral alterations such as increased anxiety [9,10]. There are demonstrated sex-based differences in HFD-induced anxiety-like behavior and sensitivity to mild, chronic stress [11]. Our lab has shown that HFD-induced changes to dopamine-related behavior and dopamine release and reuptake in the striatum and NAc were different based on sex and strain in two validated models of diet-induced obesity [12]. However, the mechanisms that contribute to HFD-induced alterations to dopamine neurochemistry and associated behavior change are poorly understood. Further, the extent to which gene–hormone interactions influence these mechanisms as well as the dopaminergic and behavioral response to an HFD are not well characterized. 

HFD-induced insulin resistance has been shown to contribute, mechanistically, to alterations in dopamine neurochemistry and associated behavior. Insulin has been demonstrated to attenuate dopamine signaling in both human subjects and animal models [13,14]. Recent evidence has confirmed that insulin enhances the DAT maximal rate of reuptake in rodent models; however, this sensitizing effect of insulin on DAT function has been shown to be reduced by an HFD [1,15]. Genetic background and biological sex have been demonstrated to influence the development of insulin resistance due to an HFD [16,17]. Despite this, the impact of strain and sex on insulin-dependent changes at the dopamine terminal remains elusive. 

Changes to brain insulin signaling is one mechanism demonstrated to lead to HFD-induced alterations to the dopaminergic system. Emerging evidence suggests that the polarization and recruitment of microglia due to an HFD is associated with changes to dopamine signaling and reuptake. An HFD has been demonstrated to initiate the polarization of microglia to an M1-like, pro-inflammatory phenotype [18,19]. Microglia sense enteric saturated fat and initiate a cascade of neuroinflammation, transcription, and the recruitment of additional M1 microglia in dopaminergic brain regions, leading to neuronal stress, increased localized pro-inflammatory cytokines, and ultimately, dopaminergic neuron degeneration [19,20,21]. It has recently been demonstrated, through chemogenetic manipulation of microglia expression in the dorsal striatum, that microglial activation leads to a reduction in the excitatory capacity of dopaminergic neurons [22]. Microglia-mediated changes to dopamine signaling have also been implicated in dopamine-related behaviors. The silencing of microglial signaling in response to an HFD and the depletion of microglia have both prevented HFD-induced weight gain and hyperphagia [23]. While sex-based differences in HFD-induced microglial activation have been identified, the influence of strain and sex on HFD-induced microglia polarization and related alterations to dopamine neurochemistry remains poorly understood [24]. 

Balb/c mice have been found to be resistant to the development of insulin resistance and the inflammatory phenotype typical of B6J strains due to an HFD [25,26,27]. Additionally, HFD-induced weight gain and visceral white adipose tissue gain are less pronounced in Balb/c mice compared to B6J counterparts [25,26]. Given its invulnerability to HFD-induced insulin resistance and a differential inflammatory response to an HFD, the Balb/c model is a powerful tool for differentiating how gene–hormone interactions influence the inflammation- and insulin-dependent mechanisms that alter dopamine neurochemistry and behavior due to HFD.

The purpose of this study was to determine the influence of sex and strain on dopamine neurochemistry and dopamine-related behavior due to a high-fat diet. B6J and Balb/c strains were chosen due to their responder and non-responder inflammatory phenotypes, respectively, when exposed to an HFD. We sought to elucidate the extent to which dopamine neurochemistry coincided with microglial activation and alterations to glucose metabolism amongst strain and sex groups. We hypothesized that high-fat diet-induced changes to dopamine neurochemistry and anxiety-like behavior will be more evident in the B6J strain, concurrent with microglial activation and metabolic dysfunction.

## 2. Materials and Methods

### 2.1. Animals and Diet

A total of 96 male and female mice from the B6J and Balb/c strains were purchased from the Jackson Laboratory (Bar Harbor, ME, USA) at post-natal day (PND) 21. Following a three-day acclimation period, the mice were randomized to either a purified control-fat diet (CFD) containing 10% kcal fat/g (D12450B; Research Diets) or a mineral-matched purified HFD containing 60% kcal fat/g (D12492; Research Diets) for 12 weeks (Figure 1). The nutrient compositions of the assigned diets are provided in Appendix A. The treatment groups are as follows: B6J male CFD, B6J male HFD, B6J female CFD, B6J female HFD, Balb/c male CFD, Balb/c male HFD, Balb/c female CFD, and Balb/c female HFD (n = 12 per group). Minimum micronutrient requirements for rodents are met with the mineral formulation (S10026; Research Diets) in both control and high-fat diets [28]. Ad libitum feeding of the randomly assigned diet and 24 h/day access to deionized water were provided for the duration of the 12-week study. Mice were housed three per cage, with ear notching utilized to identify the mouse number. Male and female cages were positioned separately, and the mice were maintained on a 12 h light/dark cycle, in a 25 ± 1 °C temperature-controlled room. Baseline weight data were collected on the day of randomization to CFD or HFD treatment and weekly, thereafter, for the duration of the study. Food intake data were collected twice weekly during the 12-week study. Behavior experiments and metabolic tests were conducted during treatment weeks 10 and 11, respectively. At the end of the 12-week dietary treatment, the mice were humanely euthanized in a manner specific to the experimental requirements for ex vivo fast scan cyclic voltammetry (FSCV) or immunohistochemistry. Mice used for FSCV were euthanized in groups of one to two daily, following a Latin square pattern, at 0900 to control for circadian alterations to dopamine neurochemistry. Therefore, the mice were divided into two cohorts (n = 48 in each cohort). Cohort 2’s arrival, acclimation, and initiation of dietary randomization occurred 3 weeks following those of cohort 1. The 3Rs principles of ethical animal research were adhered to. All protocols were accepted by the University of North Carolina at Greensboro Institutional Animal Care and Use Committee (protocol #2022-1191) and were conducted in accordance with the National Institutes of Health guidelines for the ethical care and use of laboratory animals.

### 2.2. Elevated Plus Maze

The impact of sex, strain, and diet on anxiety-like behavior was measured using validated elevated plus maze (EPM) experiments (n = 94), adapted from Walf and Frye (2007) and Pawlak et al. (2012) [29,30]. Testing was conducted between 0900 and 1000 during week 10 of the CFD or HFD treatment. The EPM apparatus is 60 cm in height, with a center square intersection (10 × 10 cm), two open opposing arms (10 × 32.5 cm), and two closed opposing arms (10 × 32.5 × 15 cm) with 15 cm high walls in a plus design. The mice were placed in the center square intersection facing the corner between the open and closed arms. Behavior in the maze was recorded for 5 min using HomeCageScan version 3.00 software (CleverSys, Inc., Reston, VA, USA) and analyzed using TopScanLite version 2.00 (CleverSys, Inc., Reston, VA, USA). The EPM apparatus was thoroughly cleaned with Quatricide disinfectant and allowed to dry between tests. The number of entrances, percent duration of time spent, distance travelled in open arms, latency, and velocity in the closed arms, open arm, and square intersection were measured. Entrance into each section of the maze was defined as the center of the body in either the center, closed arms, or open arms. 

### 2.3. Intraperitoneal Glucose Tolerance Test 

To determine the effects of diet, sex, and strain on the rate of glucose clearance, intraperitoneal glucose tolerance tests (IPGTT) were conducted during week 11 of the 12-week diet administration (n = 94). IPGTT measures glucose uptake and is an indirect measure of insulin activity, providing information about total glucose tolerance [31,32]. The mice were fasted for 15 h starting at 1700. At 0800, fasting blood glucose measurements were taken from the tail vein using a CVS Health™ Advanced Bluetooth Glucose Meter and Advanced Glucose Meter test strips (CVS Pharmacy, Woonsocket, RI, USA). IP injections of 2 g/kg glucose in 20% saline were delivered, and blood glucose measurements were taken at 15-, 30-, 60-, and 120-min following IP glucose injections. The clearance of blood glucose over time is expressed as the area under the curve (AUC). 

### 2.4. Ex Vivo Fast Scan Cyclic Voltammetry

Dopamine release and reuptake in the nucleus accumbens (NAc) core and dorsal striatum were determined by FSCV. At 0900, the mice (n = 32) were anesthetized with isoflurane and euthanized by rapid decapitation. The brains were removed and hemisected on the sagittal plane. The left hemisphere was immediately submerged in ice-cold artificial cerebrospinal fluid (aCSF; containing NaCl 126 mM, NaHCO_3_ 25 mM, D-glucose 11 mM, KCl 2.5 mM, CaCl_2_ 2.4 mM, MgCl_2_ 1.2 mM, NaH_2_PO_4_ 1.2 mM, L-ascorbic acid 0.4 mM, pH adjusted to 7.4) and coronally sliced to 300 µM with a compresstome (Precisionary Instruments; Greenville, NC, USA). Slices containing the NAc and striatum (+1.54 to +1.10 mm from bregma) were allowed to equilibrate for 60 min in oxygenated (95% O_2_/5% CO_2_) aCSF flowing at a rate of 2 mL/min at 37 °C. The scanning voltage (−0.4 to +1.2 and back to −0.4 V vs. Ag/AgCl at 400 V/s every 100 msec) was applied to a glass capillary tube-encased carbon fiber electrode (Goodfellow, C005722; Huntingdon, UK) placed approximately 75 µM deep into first the NAc core locations and then the striatal locations. For each regional placement, baseline dopamine release was evoked from a bipolar stimulating electrode (Plastics One, Roanoke, VA, USA, 8IMS3033SPCE) with a 20 Hz single pulse (1p) stimulation (mimicking tonic dopamine firing) every 300 s until peak dopamine release was within 5% variability for three consecutive readings. Once baseline dopamine release was established, phasic dopamine was represented by five serial 20 Hz firings (5p20Hz) in the same location. Baseline 1p was re-established following 5p20Hz stimulation prior to adding amphetamine (AMPH) to the aCSF to pharmacologically challenge the terminals. 

Following pre-drug recordings in the striatum, increasing AMPH concentrations (30 nM, 300 nM, 3 μM) were cumulatively applied to each slice. 1p stimulations were repeated until evoked dopamine levels reached stability at each concentration. A single 5p20Hz measurement was taken at the 3 μM AMPH concentration following 1p recordings. 

Demon Voltammetry and Analysis software was used to analyze the FSCV data. The maximal rate of dopamine uptake (Vmax), peak dopamine release, and AMPH-induced uptake inhibition (Km) were modeled with Michaelis–Menten parameters [33]. For pre-drug modeling, the baseline Km was set to 160 nM based on the affinity of dopamine for the DAT in rodent models, invariable across biological sex and diet variables. Vmax values were allowed to vary, depending on the rate of dopamine uptake [34]. Following AMPH application, Vmax was fixed at baseline values, while Km was adjusted to model changes in dopamine uptake inhibition [35]. Recording electrodes were calibrated following the recordings for each slice in order to convert electrical current measures to the evoked dopamine concentration.

### 2.5. Transcardiac Perfusion and Fixation

The male and female Balb/c and B6J mice used for immunohistochemistry (n = 16) experiments were transcardially perfused for mouse brain fixation and dissection. The mice were anesthetized with an intraperitoneal injection of 100 mg/kg ketamine in combination with 10 mg/kg xylazine. Transcardiac perfusion with 0.01 M phosphate-buffered saline (PBS) was followed by 4% paraformaldehyde in 0.01 M PBS for fixation. The brains were dissected and placed in 4% paraformaldehyde in 0.01 M PBS for 24 h, followed by 20% sucrose in 0.01 M PBS for 48 h. Perfused, fixed, and dehydrated brains were then blocked on dry ice and stored at −80 °C until cryosectioning. 

### 2.6. Immunohistochemistry

To determine the influence of strain, sex, and diet on microglial activation and its relationship to dopamine neurochemistry, immunohistochemistry was performed to identify the relative fluorescent density of protein markers of microglial activation, Iba1 and CD11b, and tyrosine hydroxylase (TH), the rate-limiting enzyme for dopamine synthesis. Frozen brains (n = 16) were cryosectioned into 30 uM thick coronal slices containing the striatum (+1.70 to +1.10 mm from bregma) using a cryostat and floated in 0.01 M PBS. Following slicing, non-specific protein blocking was accomplished by bathing the slices in 0.01 M PBS containing 0.4% Triton-X and 0.1% BSA (Bovine Serum Albumin) for 4 h on an orbital rotator. The sections were then incubated overnight and shielded from light in a primary antibody cocktail containing: anti-tyrosine hydroxylase conjugated to Alexa 488 (1:1000, catalog #: bs-0016R-A488, Bioss, Woburn, MA, USA), anti-Iba1 conjugated to Alexa Fluor 568 (1:500, catalog #: ab221003, Abcam, Cambridge, UK), and anti-CD11b conjugated to Alexa Fluor 647 (1:500, catalog #: ab197702, Abcam) in 0.01% BSA in 0.01 M PBS. Brains from the female mice were incubated in the same antibody cocktail, while male mice were incubated in a separate antibody cocktail, both following the abovementioned protocol. The slices were removed from the antibody cocktail and washed four times with 0.01 M PBS. The slices were then floated onto glass slides, mounted with 2 µL of Antifade Mounting Medium with DAPI (Vector Labs, Newark, CA, USA), and cover-slipped. The slides were stored at 4 °C until imaging. ImageJ was used to measure the relative intensity of immunofluorescence, expressed as a ratio of the signal to the background intensity. Statistical comparisons were made between the strain and diet groups but not between sexes. The expression of TH, Iba1, and CD11b was normalized to DAPI. Colocalization analysis was conducted using JACoP to calculate the Pearson Correlation Coefficient [36]. 

### 2.7. Statistical Analysis

Three-factor analysis of variance (ANOVA) was used to determine the effect of diet, sex, and strain on total grams of food eaten, IPGTT AUC, behavioral data, and dopamine kinetics. To determine the impact of sex, strain, and diet on weight over the duration of the study, blood glucose throughout IPGTT, and AMPH-induced dopamine vesicular depletion and inhibition of dopamine reuptake over the drug curve, a three-factor repeated-measures ANOVA was used. The impact of strain and diet on IHC data amongst male and female mice was determined by two-way ANOVA. For all ANOVA tests, a Bonferroni adjustment was applied to pairwise comparisons. Statistically significant main effects are reported for data in which no statistically significant interactions were determined. The normality of the data was confirmed using the Shapiro–Wilk test. For three-factor ANOVA, homogeneity of variance was determined using Levene’s test. If homogeneity of variances was not achieved to perform three-factor ANOVA, differences between the groups were assessed using two-way independent t-tests. For three-factor repeated-measures ANOVA, sphericity was confirmed with a Greenhouse–Geisser test. Linear regression analysis was conducted to determine the percent of variance in dopamine release and reuptake explained by glucose intolerance (R^2^). Group data are reported as the mean ± SEM. Statistical analysis was conducted using IBM SPSS Statistics 26. 

## 3. Results

### 3.1. Strain and Sex Influenced Weight Gain and Glucose Clearance Following 12 Weeks of an HFD

Baseline body weights were similar between the CFD and HFD groups. There was a two-way interaction between strain and diet (F_1,86_ = 24.314, *p* < 0.001). Specifically, by the end of the 12-week diet intervention, the HFD groups of all strain and sex combinations had gained significantly more weight than their CFD counterparts (*p* < 0.001 for all groups except Balb/c females *p* = 0.008) (Figure 2A,B, respectively), and HFD-fed B6J mice gained more weight, on average, than their Balb/c counterparts (*p* < 0.001 for both comparisons). There were no differences in grams of diet eaten between the CFD and HFD groups amongst male (*t*(6) = 6, *p* = 0.133) and female (*t*(4) = 1.067, *p* = 0.346) B6J mice. Comparatively, the Balb/c male (*t*(6) = −6.715, *p* = 0.007) and female (*t*(3) = −9.910, *p* = 0.007) CFD groups consumed significantly more grams of diet than the HFD counterparts. Fasting blood glucose was significantly elevated due to an HFD for B6J males (*t*(17.144) = −5.167, *p* < 0.001) and B6J females (*t*(19) = −4.494, *p* < 0.001) but not for Balb/c mice (Figure 2C). We confirmed that an HFD induced a significant increase in AUC for B6J males (*p* < 0.001), B6J females (*p* < 0.001), Balb/c males (*p* = 0.028), and Balb/c females (*p* = 0.024), indicating HFD-induced metabolic impairment (Figure 2E,F, respectively). However, the extent to which an HFD impacted glucose tolerance was differential between the two strains. The AUC was significantly lower in the HFD Balb/c males (*p* < 0.001) and females (*p* < 0.001) compared to the B6J counterparts fed an HFD, demonstrating that Balb/c mice were more resistant to HFD-induced changes in glucose tolerance compared to B6J mice (Figure 2D). B6J males fed an HFD developed more pronounced metabolic impairment compared to HFD-fed B6J females (*p* = 0.007), but no such impact of sex was seen in the Balb/c strain.

### 3.2. HFD Influenced Anxiety-like Behavior for the B6J Strain

There were no significant interactions between strain, sex, and diet on the percent duration nor bouts in the closed and open arms of the EPM (Appendix A). However, HFD-fed B6J males (*t*(20.36) = −3.089, *p* = 0.006) and females (*t*(19) = −2.917, *p* = 0.009) spent 42.36% and 37.1% more time in the center compared to the CFD counterparts, respectively (Figure 3A). B6J females fed an HFD traveled significantly slower through the center than B6J females fed a CFD (*t*(19) = 2.168, *p* = 0.043) (Figure 3E). Additionally, B6J males fed an HFD were 184% more latent in the closed arm compared with the CFD-fed counterparts, although this was not statistically significant (Figure 3B). B6J females fed an HFD travelled significantly less distance in the closed arm compared to male B6J mice fed an HFD (F_1,86_ = 3.998, *p* = 0.049), demonstrating a sex-based difference in HFD-induced anxiety-like behavior in this strain. We also demonstrated differential behavior based on the strain. B6J males and females fed a CFD or an HFD travelled significantly more distance (center (F_1,86_ = 68.733, *p* < 0.001), open arm (F_1,86_ = 49.554, *p* < 0.001) (Figure 3C), closed arm (F_1,86_ = 68.345, *p* < 0.001) (Figure 3D)), travelled faster (open arm (F_1,86_ = 80.314, *p* < 0.001)), and explored more (bouts in the center (F_1,86_ = 33.512, *p* < 0.001) (Figure 3F) and open arm (F_1,86_ = 9.757, *p* = 0.002)) compared to the Balb/c counterparts. Movement trajectory maps in the EPM are provided in Appendix A. 

### 3.3. Impact of Strain, Sex, and Diet on Dopamine Release and Reuptake

Dopamine reuptake in the NAc (F_1,64_ = 6.055, *p* = 0.017) and phasic dopamine release in the striatum (F_1,41_ = 7.036, *p* = 0.011), expressed as a percent of the baseline, were significantly reduced due to an HFD for B6J females (Figure 4A,D, respectively). Specifically, striatal dopamine release was reduced by 33% of the baseline due to an HFD for B6J females. In contrast, the mean difference between Balb/c mice fed a CFD compared to those fed an HFD was 0.270 ± 12.719 and 0.352 ± 11.555, for males and females, respectively. We also identified sex-based differences in NAc dopamine reuptake and dopamine release in the striatum. B6J females fed a CFD had significantly higher dopamine reuptake in the NAc compared to the male counterparts (F_1,64_ = 4.302, *p* = 0.042). B6J females experienced a more marked blunting of striatal dopamine release due to an HFD compared to B6J males (F_1,41_ = 8.241, *p* = 0.006). We found a negative correlation between NAc dopamine release and glucose clearance for B6J (R^2^ = 0.612, *p* = 0.002) and Balb/c (R^2^ = 0.516, *p* = 0.008) males fed an HFD (Figure 4E). The striatal dopamine reuptake rate was also strongly negatively correlated with AUC for B6J males fed an HFD (R^2^ = 0.766, *p* = 0.01) (Figure 4F). 

### 3.4. The Potency of AMPH-Induced Dopamine Uptake Inhibition and the Terminal Depletion of Dopamine Are Dependent upon Strain, Sex, and Diet

There was a significant two-way interaction between strain and sex for dopamine release, expressed as a percent of the baseline, over the duration of exposure to AMPH (F_1,34_ = 7.76, *p* = 0.009). Specifically, Balb/c females fed a CFD (F_1,34_ = 7.395, *p* = 0.01) and an HFD (F_1,34_ = 5.126, *p* = 0.03) experienced significantly less AMPH-induced dopamine depletion compared to the male counterparts (Figure 5A,B). For the CFD groups, this difference persisted through the 90 min drug curve (Figure 5A). For Balb/c mice fed an HFD, this significant sex-based difference emerged at the 30 nm dose, persisted into the 300 nm dose, and normalized with 3 uM of AMPH (Figure 5B). Additionally, Balb/c males fed an HFD experienced a significantly greater AMPH-induced depletion of dopamine compared to B6J males fed an HFD (F_1,34_ = 4.124, *p* = 0.05) (Figure 5C). B6J males were the only biological group to experience differences in AMPH-induced dopamine depletion due to diet (Figure 5D). At the 30 nm AMPH dose, B6J males fed an HFD experienced a significantly greater increase in dopamine release from the baseline compared with CFD-fed B6J males (F_1,34_ = 5.144, *p* = 0.03). This difference improved with greater doses of AMPH. 

At the 30 nm dose of AMPH, Balb/c males fed an HFD experienced greater AMPH potency, measured by the greater uptake inhibition (Km) at a lower dose of AMPH, compared to B6J males fed an HFD at both 15 (F_1,33_ = 7.148, *p* = 0.012) and 30 min (F_1,33_ = 6.576, *p* = 0.015) (Figure 5E). Similarly, Balb/c males fed an HFD experienced a greater AMPH potency within 15 min (F_1,33_ = 6.245, *p* = 0.018) of this dose compared to CFD-fed counterparts, which was sustained until 30 min (F_1,33_ = 5.829, *p* = 0.021) (Figure 5F). These differences normalized within the 300 nm dose of AMPH. Additionally, Balb/c males fed an HFD experienced a significantly greater AMPH potency within 30 min of the 30 nm dose compared to Balb/c females fed an HFD (F_1,33_ = 7.178, *p* = 0.011) (Figure 5G). Comparisons of dopamine release over the AMPH curve and AMPH potency between CFD- and HFD-fed mice for the remaining biological groups are provided in Appendix A. 

### 3.5. Influence of Strain and Sex on HFD-Induced Microglial Activation and TH Expression

TH expression was increased for B6J males (*t*(102) = −4.764, *p* < 0.001) but reduced for Balb/c males (*t*(87.829) = 2.105, *p* = 0.038) due to an HFD (Figure 6B). There was a significant two-way interaction between strain and diet for CD11b expression (F_1,172_ = 25.311, *p* < 0.001). Specifically, CD11b expression was significantly increased due to diet for both Balb/c (F_1,172_ = 12.916, *p* < 0.001) and B6J males (F_1,172_ = 12.429, *p* < 0.001). However, the expression was significantly higher in B6J males fed an HFD compared to Balb/c counterparts (F_1,172_ = 44.046, *p* < 0.001) (Figure 6C). The expression of Iba1 was also significantly increased by 25.7% due to an HFD for B6J males (*t*(93.545) = −9.102, *p* < 0.001) but not Balb/c males (Figure 6D). Additionally, the colocalization of Iba1 and CD11b was increased due to an HFD for B6J males (*t*(96) = 21.219, *p* < 0.001) but not Balb/c males. These data suggest that the microglia that were recruited due to an HFD became activated amongst B6J males. Similar to B6J males, TH expression was increased due to an HFD for B6J females (F_1,219_ = 74.162, *p* < 0.001). Balb/c females experienced a similar trend in TH expression due to an HFD, but the expression was significantly lower than that of B6J females (F_1,219_ = 41.293, *p* < 0.001) (Figure 7B). There were no significant changes in CD11b expression for B6J and Balb/c females due to diet (Figure 7C). However, the expression of IBA1 for B6J females fed an HFD was significantly higher than the Iba1 expression of both B6J CFD-fed females (F_1,218_ = 31.591, *p* < 0.001) and Balb/c HFD-fed counterparts (F_1,218_ = 30.119, *p* < 0.001) (Figure 7D). Specifically, the expression of Iba1 was increased by 18.3% due to diet for B6J females. Additionally, Iba1 and CD11b colocalization was increased due to an HFD for B6J females (*t*(109) = 3.135, *p* = 0.002) but reduced for Balb/c females (*t*(57.488) = −5.140, *p* < 0.001). 

## 4. Discussion

The purpose of our study was to determine the influence of sex and strain on HFD-induced insulin resistance, microglial activation, dopamine kinetics, and associated behavior using male and female B6J and Balb/c mice. The Balb/c strain is resilient to the HFD-induced reduction in glucose clearance, adiposity, and immune cell infiltration, which allowed us to determine the impact of sex and strain on changes in dopamine kinetics and associated behavior due to an HFD. Overall, our findings show that genetic background and sex influence the dopaminergic and behavioral changes due to an HFD that arise from mechanisms related to metabolic impairment and microglial activation. Notably, when fed an HFD, mice of the B6J strain developed more pronounced glucose dysregulation than Balb/c mice, female B6J mice experienced reduced striatal dopamine release and NAc dopamine reuptake, variance in dopamine kinetics was explained by differences in glucose clearance only for males, B6J males experienced an increase in indicators of microglial activation, and B6J mice spent more time in the center of the EPM. These findings support our hypothesis that HFD-induced microglial activation, insulin resistance, and associated changes to dopamine kinetics and behavior would have a greater impact on B6J mice. Balb/c males demonstrated increased AMPH-induced vesicular dopamine depletion at the terminals in the striatum and reuptake inhibition, further corroborating our hypothesis that Balb/c and B6J mice would have discrete responses to an HFD. 

The B6J strain is a validated model of DIO and develops insulin resistance when fed an HFD. It is well characterized that Balb/c mice do not develop the same HFD-induced alterations to glucose homeostasis compared to the B6J strain [25]. We reproduced a metabolic syndrome-like phenotype in B6J males and females fed an HFD characterized by increased fasting blood glucose and insulin resistance, as demonstrated by IPGTT. There was no change in fasting blood glucose due to diet within the Balb/c strain. Despite this, we found that glucose clearance was significantly impaired for Balb/c males and females due to an HFD, although this metabolic dysfunction amongst HFD-fed B6J mice was more pronounced compared to that of HFD-fed Balb/c mice. Insulin signaling enhances dopamine release and reuptake via DAT in the striatum, playing a role in nutrient sensing and food intake through actions on motivation and reward circuits [15]. We found that impaired glucose clearance was negatively associated with NAc dopamine release for HFD-fed Balb/c and B6J males. Diminished glucose clearance explained more of the variation in NAc dopamine release for B6J males as compared to Balb/c males. We also demonstrated that metabolic dysfunction was strongly, negatively correlated with and explained 76.6% of the variation in striatal dopamine reuptake for B6J males fed a high-fat diet. A previous study from Fordahl & Jones (2017) utilizing B6J males fed a 60% HFD for 6 weeks showed that dopamine reuptake in the NAc was negatively correlated with insulin resistance, but there was no relationship found between NAc dopamine release and insulin resistance [1]. Our findings support insulin resistance as a mechanism for HFD-induced alterations at the dopamine terminal in the striatum and NAc for males, particularly of the B6J background. The lack of a relationship between alterations to glucose metabolism and the HFD-induced reductions in striatal dopamine release and NAc reuptake in B6J females suggests an alternative mechanism for females. 

We demonstrated that striatal dopamine release was reduced by 33% of the baseline due to an HFD for B6J females. Interestingly, our lab previously found that B6J females experienced a 24% increase in striatal dopamine release following 16 weeks of an HFD [12]. Our results showed that dopamine release in this brain region was increased due to am HFD for B6J males, but this was not significant. These data suggest that striatal dopamine release for B6J females is highly impacted by an HFD, and the response to an HFD is differentially influenced by the duration of exposure. Several studies, ranging from 4 to 12 weeks of exposure to obesogenic, high-fat feedings, demonstrated that striatal dopamine release and reuptake is reduced in HFD-fed male and female rats [15,37,38]. However, consistent with the trend in our lab’s previous findings within B6J females, longer-term (16 weeks) access to an HFD has been found to increase dopamine release in the dorsal striatum for B6J males [39]. It has been demonstrated that striatal dopamine release and DAT reuptake are greater amongst mice compared to rats; however, it remains unclear whether there is a differential response to an HFD between rat and mouse models or whether the temporal differences in exposure to an HFD explain the variation in HFD-induced changes in striatal dopamine release [15]. 

In the NAc, we found that dopamine reuptake was decreased due to an HFD for B6J females. This trend is consistent with several studies, using male mouse and rat models, that demonstrated NAc dopamine reuptake was decreased due to high-fat feedings [1,15,40,41]. We also demonstrated sex-based differences in B6J mice fed a CFD, which was not congruent with our lab’s previous findings following 16 weeks of a CFD or HFD [12]. It had been shown that dopamine reuptake in the striatum is greater for female rats compared to males and is thought to be due to the greater auto receptor control of DAT amongst females [42,43]. While sex-based differences in the expression of NAc and striatal dopamine receptors D1 and D2 have been reported, sex-based differences in DAT expression in the NAc have yet to be elucidated [44,45]. However, these differences in the dopaminergic system in males and females have been implicated in the sex-based diversity of dopamine-related behavior and neuropsychiatric disorders [46]. 

The HFD-induced alterations to B6J dopamine release in the striatum and reuptake in the NAc were not observed in Balb/c mice. There was less than a 1% difference in striatal dopamine release between CFD- and HFD-fed groups of the Balb/c strain. It has been demonstrated that Balb/c mice express lower striatal dopamine levels, determined by high-performance liquid chromatography, by post-natal day 30 as compared to B6J mice, suggesting differences in dopaminergic system development between these strains [47]. However, we found no differences in striatal dopamine release or TH expression strictly due to strain. 

Amphetamine (AMPH) increases extracellular dopamine by competitively inhibiting DAT, increasing the movement of vesicular dopamine, and facilitating the DAT-mediated reverse transport of dopamine [48]. These mechanisms prolong dopaminergic signaling to receptors in the striatum, leading to dopamine receptor saturation and, eventually, the depletion of intracellular dopamine stores [49,50]. Ex vivo models have demonstrated that AMPH decreases the affinity of dopamine for its transporter at doses of 100–300 nm, while a reduction in DA release occurs at higher doses of 1–3 µM [51,52]. We found that early in the drug curve, B6J males experienced differences in AMPH-induced striatal dopamine release due to diet. Specifically, B6J males fed an HFD experienced increased AMPH-induced dopamine release compared to CFD-fed counterparts at the 30 nm dose. This trend is consistent with findings from a study conducted in B6J males fed a 60% HFD for 6 weeks in which the AMPH-induced vesicular depletion of dopamine in the NAc was greater for control mice [40]. Alternatively, a study of male Sprague–Dawley rats fed a 60% HFD from PND 21 to PND 62 found that NAc dopamine release was unchanged from the baseline following a 100 nm dose of AMPH. We found that Balb/c females fed a CFD or HFD experienced less AMPH potency and AMPH-induced depletion of dopamine than Balb/c males. Balb/c males fed an HFD appeared to experience the greatest sensitivity to AMPH-induced dopamine uptake inhibition and the depletion of dopamine. DAT function and vulnerability to AMPH have been found to be causally linked [53]. The greater expression of DAT in the VTA and SN amongst Balb/c males when compared to B6J males may, to some degree, explain strain-based differences in AMPH potency, as this is positively correlated with DAT density [54]. Our data demonstrate that Balb/c males fed an HFD have a distinct response to AMPH-induced uptake inhibition and the vesicular depletion of dopamine, which exemplifies the impact of strain and sex in response to an HFD.

We found that tyrosine hydroxylase protein expression was significantly increased in the striatum due to diet for B6J mice and reduced for Balb/c males. Our lab has previously demonstrated that following 16 weeks of an HFD, striatal TH gene expression was significantly increased in B6J males and females. However, HFD-induced changes to TH expression may be highly brain region-specific. In one study of 12-week-old B6J males fed a 40% HFD for 8 weeks, TH gene expression was significantly reduced in several brain regions, including the hypothalamus, substantia nigra, and ventral tegmental area, compared to controls [55]. Similarly, 37-week-old male rats fed a 45% HFD for 30 weeks experienced a reduction in TH-containing neurons with a concurrent reduction in Iba1 within the substantia nigra, implicating microglial activity as a mechanism for changes to dopamine within this brain region [56]. 

One’s genetic background and biological sex influence the microglial response [57,58]. Sex-specific differences in the microglia-mediated development of the dopaminergic system have been demonstrated during postnatal and adolescent periods, impacting dopamine-related behavior [59]. The sex-based heterogeneity of microglial activity during periods of development extend into adulthood and may explain any differential response to inflammation and subsequent microglial activation [60,61]. The recruitment and activation of microglia have been demonstrated to be a consequence of an HFD in mouse models and have been observed in patients with obesity [2]. We found that Iba1 expression and colocalization with CD11b in the striatum were significantly increased due to an HFD for B6J males and females, but the effect of diet on expression was negligible for Balb/c mice. Our findings are consistent with several other studies using adult male rodent models that demonstrated increased Iba1 expression in the hypothalamus following 3 days to 8 weeks of an HFD, leading to weight gain and hyperphagia [2,18,19]. This elevation in Iba1 due to an HFD has also been reported in the prefrontal cortex of 4-month-old B6J males following 14 weeks of a 60% HFD [62]. We found that CD11b expression was increased due to diet for B6J males but decreased due to diet for Balb/c males. There was also a strain-based difference in striatal CD11b expression for females, such that B6J females fed either a CFD or an HFD had a greater expression of CD11b compared to Balb/c counterparts. An HFD has been found to increase the global central nervous system recruitment of CD11b-positive immune cells in male B6J mice [63]. One study of 12-week-old toxin-induced Parkinsons disease models from the B6J background fed a 60% HFD for 21 weeks demonstrated a decrease in the interaction between CD11b-positive cells and blood vessels of the striatum, demonstrating that HFD-induced changes to the function of activated microglia may play a role in the pathogenesis of dopaminergic degeneration [64]. Estrogen has been demonstrated to block microglial activation following exposure to inflammatory stimuli, which may explain the sex-based divergence in response to HFD-induced CD11b expression [65].

Increased frequency in anxiety-like symptoms has been demonstrated in humans with obesity and in HFD-fed rodent models [66,67,68]. We found that HFD-fed B6J males and females spent significantly more time in the center of the maze, where they were placed, than CFD-fed Balb/c males and females. We did not find that there were differences in the time spent or bouts in open or closed arms due to diet amongst any biological group. Although, we did find that B6J males were 184% more latent in the closed arm due to an HFD, which may suggest avoidance-like or respite-seeking behavior. Our findings are consistent with a recent study of B6J males fed a 60% HFD for 7 weeks starting at 7 weeks of age, which revealed no differences in the time spent in open or closed arms due to an HFD [69]. However, another study of 2-month-old B6J males fed a 60% HFD for 4 months reported decreased time spent in the center of open field tests and in the open arm of the EPM tests, indicating increased anxiety-like behavior due to an HFD [66]. These studies suggest that anxiety-like behavior change due to an HFD is revealed over longer durations of am HFD. Interestingly, increased anxiety-like behavior measured by EPM and open field tests have been associated with the HFD-induced activation of microglia in B6J males, even following shorter durations (8 weeks) of an HFD [70]. We also found differences in behavior due to the strain. B6J mice traveled more distance in each area of the maze and traveled faster in the open arm as compared to Balb/c mice, which are known to exhibit increased anxiety-like behavior in the stress-induced hyperthermia paradigm and behavioral light–dark exploration tests compared to B6J mice. During light–dark exploration tests, it has been observed that Balb/c mice often did not move for long durations of time, even within the light area [71]. Balb/c mice have also been known to exhibit novel peeping behaviors at high frequencies compared to other strains [72]. We observed similar stagnation and peeping behaviors in both male and female Balb/c mice described in previous reports. B6J females have been demonstrated to spend more time in the open arm and make more transitions through the EPM as compared to Balb/c females [73]. Consistent with these reports, we found that B6J mice had significantly more bouts into the center and open arms compared to the Balb/c strain. One limitation of our study includes the lack of additional behavioral tests for studying anxiety, such as an open field, light–dark box assay. Additionally, the significant difference in the exploration of the maze between Balb/c and B6J mice makes it difficult to interpret the results of the EPM experiments. Future research should consider the inclusion of several approach-avoidant behavior tests in order to better evaluate anxiety-like behavior. 

## 5. Conclusions

In conclusion, our novel data show that the insulin- and microglial-dependent mechanisms that lead to alterations in dopamine neurochemistry and related behavior due to an HFD are greatly influenced by strain and sex. Importantly, we demonstrated that the variation in HFD-induced dopamine release and reuptake in the striatum and NAc was explained by insulin resistance for B6J males and, to a lesser extent, for Balb/c males, suggesting different mechanisms are responsible for HFD-induced changes at the dopamine terminal for females. Notably, Balb/c males demonstrated differential AMPH-induced deficits to the dopaminergic system and signs of reduced microglial activation due to an HFD that were largely absent in Balb/c females. Our results show that alterations to glucose homeostasis and microglial function due to an HFD contribute to changes in dopamine neurochemistry and are highly impacted by strain–sex interactions. The variability of a response to an HFD due to gene–hormone interactions may be key to understanding the differential environmental impact on the risk and pathophysiology of disorders involving perturbations to dopamine neurochemistry, like obesity. Future research will use these data to continue to explore the strain- and sex-specific mechanisms of altered dopamine homeostasis in the context of an HFD. Further determination of these mechanisms will provide insight into the complex interactions between genetics, hormones, diet, and neurochemistry in the setting of chronic disease development. 

## Figures and Tables

**Figure 1 nutrients-16-03301-f001:**
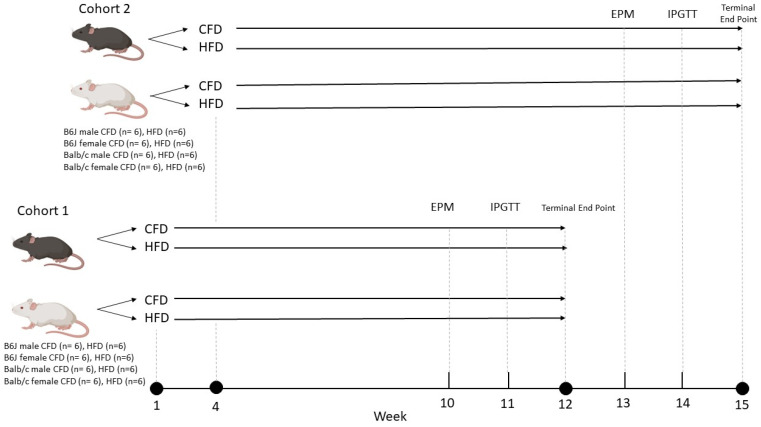
Study design schematic. A total of 96 male and female B6J and Balb/c mice aged post-natal day 21 were divided into two cohorts (n = 48 in each cohort). Following a three-day acclimation period, the mice were randomized to either a purified control-fat diet (CFD) containing 10% kcal fat/g or a mineral-matched purified HFD containing 60% kcal fat/g for 12 weeks. Cohort 2’s arrival, acclimation, and initiation of dietary randomization occurred 3 weeks following those of cohort 1. Two mice were removed from the study due to the identification of a humane endpoint. Elevated plus maze experiments (n = 94) and intraperitoneal glucose tolerance tests (n = 94) were conducted at weeks 10 and 11, respectively. Mice were humanely sacrificed at week 12. Ex vivo fast scan cyclic voltammetry (n = 32) and immunohistochemistry (n = 16) experiments were conducted to determine the influence of strain, sex, and diet on dopamine kinetics and microglial activation.

**Figure 2 nutrients-16-03301-f002:**
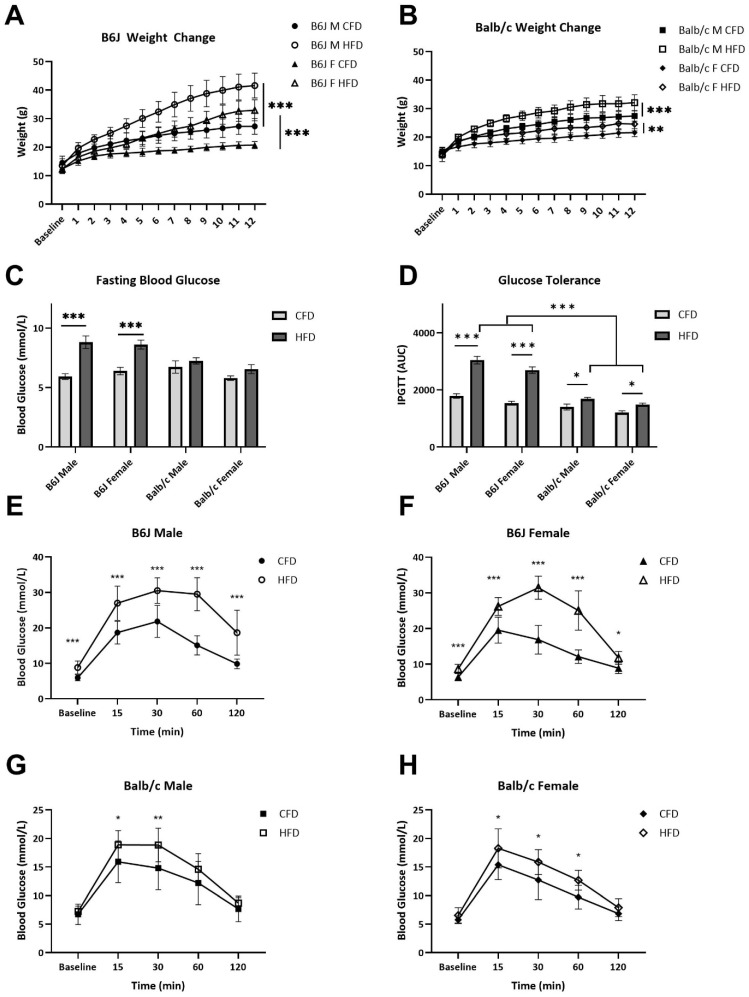
Effect of strain, sex, and diet on weight gain and insulin resistance. Weights were taken at baseline and then weekly for the duration of the 12-week study. B6J (**A**) and Balb/c (**B**) males and females fed a high-fat diet (HFD) gained significantly more weight than control-fat diet-fed counterparts. Although, the magnitude of difference between Balb/c CFD and HFD groups was lower than that of B6J groups. At week 11 of the 12-week study, IPGTT was performed to provide an indirect measurement of insulin resistance. Following 15 h of fasting, baseline blood glucose measurements were significantly elevated for B6J males and females fed an HFD (**C**). Intraperitoneal injections of 2 g/kg of glucose in 20% saline were delivered, blood glucose measurements were taken at 15-, 30-, 60-, and 90-min post-injection, and the area under the curve (AUC) was calculated (**D**). Blood glucose over the duration of the IPGTT is plotted for B6J males (**E**), B6J females (**F**), Balb/c males (**G**), and Balb/c females (**H**). * *p* < 0.05, ** *p* < 0.01, *** *p* < 0.001; data are expressed as the mean ± SEM. CFD: control fat diet, HFD: high-fat diet, IPGTT: intraperitoneal glucose tolerance test, F: female, M: male.

**Figure 3 nutrients-16-03301-f003:**
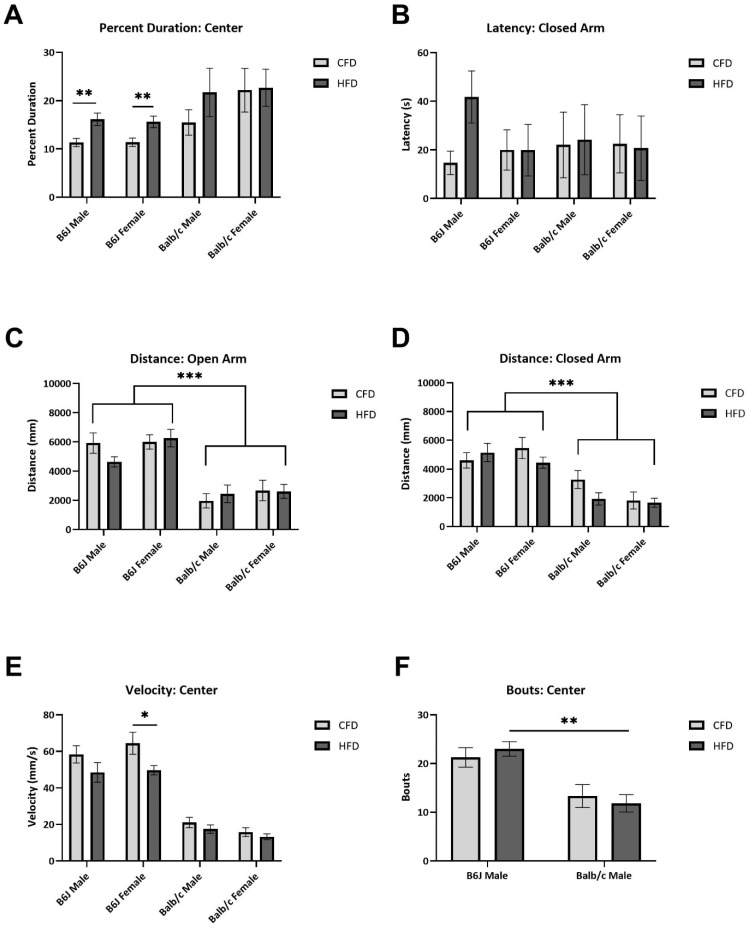
Influence of strain, sex, and diet on anxiety-like behavior. Male and female B6J and Balb/c mice (n = 94) underwent elevated plus maze behavioral tests at week 10 of the 12-week study. B6J males and females fed an HFD spent more time in the center of the maze, where they were placed, compared to CFD-fed counterparts (**A**). B6J males experienced a biologically relevant increase in closed-arm latency due to an HFD (**B**). There were demonstrated strain differences in the distance traveled in the open arm (**C**) and closed arm (**D**). B6J females moved slower through the center due to an HFD (**E**), and B6J males fed an HFD had more bouts into the center of the maze than Balb/c males fed an HFD (**F**). * *p* < 0.05, ** *p* < 0.01, *** *p* < 0.001; data are expressed as the mean ± SEM. CFD: control fat diet, HFD: high-fat diet.

**Figure 4 nutrients-16-03301-f004:**
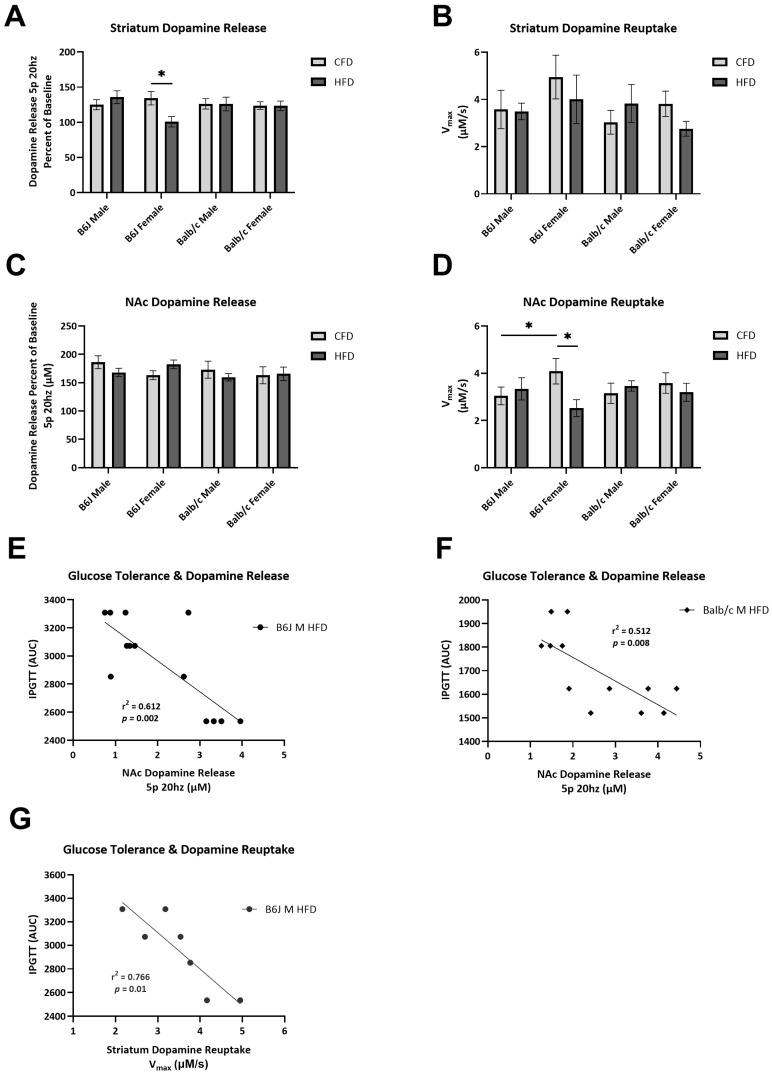
Effect of strain, sex, and diet on dopamine kinetics and the relationship to insulin resistance in the striatum and nucleus accumbens (NAc) core. Dopamine release evoked by a five-pulse stimulation at 20 Hz (“phasic” dopamine release) in the dorsal striatum (**A**) and NAc core (**C**) expressed as a percent of the baseline evoked by a single-pulse 20 Hz stimulation (“tonic” dopamine release). (**B**) Maximal rate of dopamine reuptake (Vmax) in the striatum (**B**) and NAc (**D**) following physiologically phasic stimulation. Altered blood glucose clearance, represented by area under the curve, was negatively correlated with dopamine release in the NAc core for B6J (**E**) and Balb/c (**F**) males fed a high-fat diet and dopamine reuptake in the dorsal striatum for B6J males (**G**) based on linear regression analysis. * *p* < 0.05; data are expressed as the mean ± SEM. CFD: control fat diet, HFD: high-fat diet, AUC: area under the curve F: female, M: male.

**Figure 5 nutrients-16-03301-f005:**
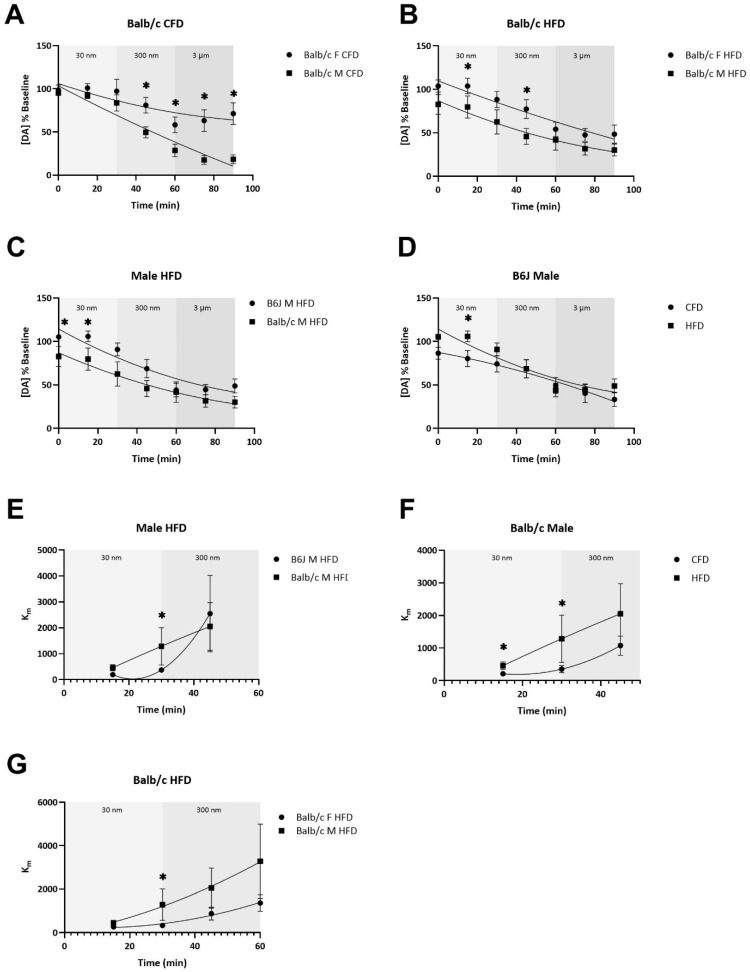
The influence of strain, sex, and diet on AMPH-induced dopamine depletion from striatal terminals and inhibition of dopamine reuptake. Striatal dopamine release and AMPH potency, or apparent Km, were measured following the application of increasing AMPH concentrations (30 nM, 300 nM, 3 μM), cumulatively applied to each slice in increments of 30 min. There were sex-based differences in the AMPH-induced depletion of dopamine amongst Balb/c mice fed either a CFD (**A**) or an HFD (**B**) and strain -based differences for males fed an HFD (**C**). Early in the drug curve, B6J males experienced differences in AMPH-induced dopamine release due to diet (**D**). Males fed an HFD experienced strain-based differences in AMPH potency within 30 min of exposure to 30 nm AMPH (**E**). Balb/c males demonstrated differences in AMPH potency due to an HFD (**F**). Balb/c females fed an HFD experienced less AMPH potency than Balb/c males (**G**). * *p* < 0.05; data are expressed as the mean ± SEM. AMPH: amphetamine, CFD: control fat diet, HFD: high-fat diet, F: female, M: male.

**Figure 6 nutrients-16-03301-f006:**
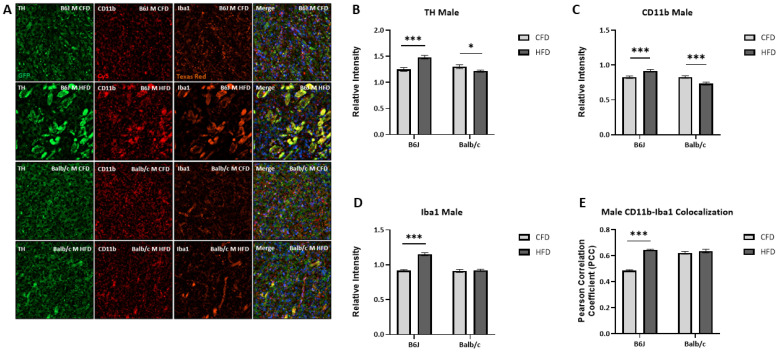
The impact of a HFD on markers of microglial activation in the striatum for B6J and Balb/c males. Immunohistochemistry was used to determine the fluorescent density of tyrosine hydroxylase (GFP, green), the rate-limiting enzyme for dopamine synthesis, and two markers of activated microglia, CD11b (Cy5, red) and Iba1 (Texas Red, rust), relative to DAPI nuclei staining in the striatum (**A**). Tyrosine hydroxylase (**B**) and CD11b (**C**) expression was altered due to diet for both B6J and Balb/c males. Iba1 expression was increased for B6J males fed an HFD (**D**). There was increased colocalization between Iba1 and CD11b due to an HFD for B6J males (**E**). * *p* < 0.05, *** *p* < 0.001; data are expressed as the mean ± SEM. CFD: control fat diet, HFD: high-fat diet, CD11b: cluster of differentiation molecule 11b, Iba1: ionized calcium binding adaptor molecule, TH: tyrosine hydroxylase.

**Figure 7 nutrients-16-03301-f007:**
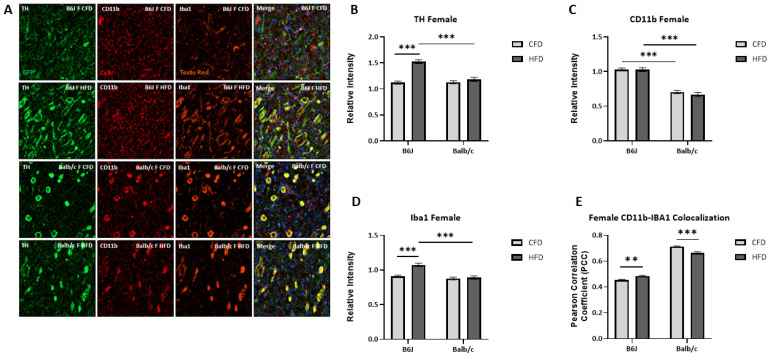
The impact of an HFD on markers of microglial activation in the striatum for B6J and Balb/c females. Immunohistochemistry was used to determine the fluorescent density of tyrosine hydroxylase (GFP, green), the rate-limiting enzyme for dopamine synthesis, and two markers of activated microglia, CD11b (Cy5, red) and Iba1 (Texas Red, rust), relative to DAPI nuclei staining in the striatum (**A**). Tyrosine hydroxylase (**B**) and Iba1 (**D**) expression was increased due to an HFD for B6J females. CD11b relative intensity was greater amongst B6J compared to Balb/c females (**C**). Colocalization between Iba1 and CD11b was increased for B6J females and reduced for Balb/c females due to diet (**E**). ** *p* < 0.01, *** *p* < 0.001; data are expressed as the mean ± SEM. CFD: control fat diet, HFD: high-fat diet, CD11b: cluster of differentiation molecule 11b, Iba1: ionized calcium binding adaptor molecule, TH: tyrosine hydroxylase.

## Data Availability

There are limited data repositories suitable for this data type. The data associated with this manuscript may be requested from the senior author, Professor Keith Erikson; kmerikso@uncg.edu.

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
