# Peer review of "The Influence of Strain and Sex on High Fat Diet-Associated Alterations of Dopamine Neurochemistry in Mice"

_nutrients, 2024, doi:10.3390/nu16193301_

Round 1

Reviewer 1 Report

Comments and Suggestions for Authors

This study assesses the impact of sex in HFD-induced glucose tolerance and anxiety-like behaviors as well as changes in dopamine and microglia activation in C57BL/6 vs Balb/c mice. The authors show that the glucose tolerance test is impaired in male and female C57 but not in Balb/c mice although changes in body weight were observed in HFD-exposed animals of both mouse strains. In terms of anxiety-like behaviors, it is better to present the time in the open and closed arms and the number of entries in the open and closed arms. If these are not different, then this should be used to interpret the anxiety-like behaviors in this study. Also, considering that C57 mice are more active than Balb/c mice, this should be considered when interpreting the results of the current study. Also, it is better to show the raw data instead of normalizing them, especially dopamine data. The correlation appears flat in Balb/c mice and this should be considered when interpreting the data.

Comments on the Quality of English Language

N/A

Author Response

The authors thank the reviewer for their comments. We have prepared our responses to the comments in the following table:

Reviewer Comment

Author Response

In terms of anxiety-like behaviors, it is better to present the time in the open and closed arms and the number of entries in the open and closed arms. If these are not different, then this should be used to interpret the anxiety-like behaviors in this study.

Also, considering that C57 mice are more active than Balb/c mice, this should be considered when interpreting the results of the current study.

We include these results in our discussion of EPM results and interpretation (lines 561-563). We agree that presenting the percent duration and bout data for open and closed arms would be of benefit to the manuscript. Therefore, we have provided a supplemental figure with this data presented (Supplementary Figure 1, lines 820-826). We have additionally added a statement in the results regarding these specific findings (lines 282-283). We have further elaborated on the limitations related to the interpretation of EPM data, including consideration of the difference in activity between strains (lines 282-283).

Also, it is better to show the raw data instead of normalizing them, especially dopamine data.

We respectfully prefer to express phasic dopamine release as a percent of tonic release or baseline as this demonstrates terminal capacity. For example, a high phasic (5p) dopamine release, expressed independent of baseline, may reflect either a high phasic response or simply a high baseline response. Expressing phasic release normalized to single pulse represents the delta between single pulse, which represents tonic pacemaker activity, and phasic burst firing that resembles physiological action potential trains. The change in synaptic dopamine from tonic to phasic firing that is critical to interpret how post-synaptic D2 or D1 receptors respond to the signal.

The correlation appears flat in Balb/c mice and this should be considered when interpreting the data.

Thank you for highlighting this. This occurred because we graphically represented two data sets on the same graph, and therefore the scale of the y axis scale was set for both data sets. We have separated each regression analysis on its own graph to better visualize the relationship between our dopamine data and glucose tolerance in Figure 4 (lines 323-333). 

Reviewer 2 Report

Comments and Suggestions for Authors

This study investigated the influence of sex and strain on striatal and nucleus accumbens dopamine neurochemistry and dopamine-related behavior due to a highly saturated fat diet (HFD). It is expected that it will provide valuable help in research on dopamine responses caused by high-fat diets.

Overall, it is considered to be a good research result in which the topic setting, research methods, and presentation of results were well organized. However, correction and supplementation of the following matters are required.

In the introduction, prior research on the influence of sex and strain is somewhat insufficient. We hope that this will supplement the need for research in this area.

Consideration of the influence of sex and strain should be added to the discussion.

The conclusion should be presented separately as a subsection.

Author Response

(The authors gave the same response as above.)

Author Response

In the introduction, prior research on the influence of sex and strain is somewhat insufficient. We hope that this will supplement the need for research in this area.

One novel aspect of our study is the comparison of strain and sex on response to a HFD. We appreciate the reviewer’s acknowledgement of the need for research in this area and agree.

Consideration of the influence of sex and strain should be added to the discussion.

We chose to organize our discussion by result and discuss sex and strain influences relevant to that specific result rather than include this discussion as an independent section (lines: 437-438, 468-471, 481-487, 490-493, 514-517, 531-536, 554-557, 575-585).  

The conclusion should be presented separately as a subsection.

We have separated the conclusion into its own section (line 591).

Reviewer 3 Report

Comments and Suggestions for Authors

The current manuscript investigates the influence of sex and strain on striatal and nucleus accumbens dopamine neurochemistry and dopamine-related behavior in response to a high saturated fat diet. The study is interesting and presents some innovative and novel aspects. However, the writing and organization of the content need further improvement.

  1. The authors should provide the nutritional composition of the feed and an analysis of the food intake.
  2. Can anxiety behavior be determined solely based on the elevated plus maze results? The evaluation of anxiety-related behaviors should be more diverse. I recommend including additional behavioral assessments or measuring anxiety-related indicators.
  3. Should the method of comparison between groups in the figures be adjusted? The current focus is only on comparing CFD and HFD groups, but statistical analysis of differences between different sexes on the same diet and different strains on the same diet has not been conducted.
  4. It would be beneficial to include behavioral trajectory maps of the animals in the behavioral analysis section.
Comments on the Quality of English Language

No

Author Response

(The authors gave the same response as above.)

Author Response

  1. The authors should provide the nutritional composition of the feed and an analysis of the food intake.

We agree with the reviewer that it would be important to add detailed information on nutritional composition of the diets used. Therefore, we have added a supplementary table with this information (referred to in line 97). Supplemental table included on lines 819-820. Our analysis of the food intake is included on (lines 252-256).

2.       Can anxiety behavior be determined solely based on the elevated plus maze results? The evaluation of anxiety-related behaviors should be more diverse. I recommend including additional behavioral assessments or measuring anxiety-related indicators.

Our current study builds on previous work by our lab (Totten et al 2022, reference 12) which included the open field test. Here, we chose the EPM as an additional modality to evaluate the impact of strain, sex, and diet on dopamine-related behavior. We agree that the evaluation of anxiety-like behavior should not be limited to one behavioral test. Therefore, we have included this as a limitation in our discussion (lines 586-591).

3.       Should the method of comparison between groups in the figures be adjusted? The current focus is only on comparing CFD and HFD groups, but statistical analysis of differences between different sexes on the same diet and different strains on the same diet has not been conducted.

We conducted a 3-way ANOVA or 3-way repeated ANOVA on our data unless statistical assumptions were violated, in which case we used a t-test to evaluate differences due to diet. Based on 3-way ANOVA results, we reported any interactions between sex, strain, and diet. Main effects were reported if there were no interactions.

4.       It would be beneficial to include behavioral trajectory maps of the animals in the behavioral analysis section.

We have provided representative behavioral trajectory maps for each strain/sex/diet group in a supplemental figure. Referred to in lines 289-299.

Round 2

Reviewer 1 Report

Comments and Suggestions for Authors

The authors adequately responded to my comments. I have no further comments.